# Fossil black smoker yields oxygen isotopic composition of Neoproterozoic seawater

F. Hodel [1,2], M. Macouin[1], R.I.F. Trindade[2], A. Triantafyllou[3], J. Ganne[1], V. Chavagnac[1], J. Berger[1], M. Rospabé [1], C. Destrigneville[1], J. Carlut[4], N. Ennih[5] & P. Agrinier[4]

The evolution of the seawater oxygen isotopic composition ($\delta^{18}O$) through geological time remains controversial. Yet, the past $\delta^{18}O_{seawater}$ is key to assess past seawater temperatures, providing insights into past climate change and life evolution. Here we provide a new and unprecedentedly precise $\delta^{18}O$ value of $-1.33 \pm 0.98‰$ for the Neoproterozoic bottom sea-water supporting a constant oxygen isotope composition through time. We demonstrate that the Aït Ahmane ultramafic unit of the ca. 760 Ma Bou Azzer ophiolite (Morocco) host a fossil black smoker-type hydrothermal system. In this system we analyzed an untapped archive for the ocean oxygen isotopic composition consisting in pure magnetite veins directly precipitated from a Neoproterozoic seawater-derived fluid. Our results suggest that, while $\delta^{18}O_{seawater}$ and submarine hydrothermal processes were likely similar to present day, Neoproterozoic oceans were 15–30 °C warmer on the eve of the Sturtian glaciation and the major life diversification that followed.

[1] Géosciences Environnement Toulouse (GET), Observatoire Midi Pyrénées, Université de Toulouse, CNRS, IRD, UPS, 31400 Toulouse, France. [2] Departamento de Geofísica, Instituto de Astronomia, Geofísica e Ciências Atmosféricas, Universidade de São Paulo, 05508-090 São Paulo, Brazil. [3] Laboratoire de Planétologie et Géodynamique, UMR-CNRS 6112, Université de Nantes, 44322 Nantes, France. [4] Institut de Physique du Globe de Paris, Université Sorbonne Paris Cité, Université Paris Diderot, CNRS, UMR 7154, 75005 Paris, France. [5] EGGPG, Département de Géologie, Faculté des Sciences, Université Chouaib Doukkali, 24000 El Jadida, Morocco. Correspondence and requests for materials should be addressed to F.H. (email: florent.hodel@hotmail.fr)

Were ancient oceans warmer than present-day oceans? This is an old and still unsettled debate that relates directly to the evolution of $\delta^{18}O_{seawater}$ through time[1-9]. On the basis of the early recognition that the older the carbonates, the more negative their $\delta^{18}O$ values, some authors suggested that $\delta^{18}O_{seawater}$ has markedly increased since the Precambrian[2,5]. Jaffrés et al.[2] modeled an evolution from −13.3 to 0‰ of the $\delta^{18}O_{seawater}$ since 3.4 Ga; the $\delta^{18}O_{seawater}$ for the Neoproterozoic being −6.4‰. According to this hypothesis, seawater temperature would be rather constant through time with temperatures similar to present-day ones since 3.4 Ga[2]. Conversely, theoretical considerations suggest that the $\delta^{18}O$ of the oceans remained buffered to a value of 0 ± 2‰ since the early Archean due to seawater interaction with the oceanic lithosphere[4,9]. Recently, this idea of a rather constant $\delta^{18}O_{seawater}$ for almost all the Earth's history was supported by data from chert-hosted kerogen[6] and serpentinite[10], which provided $\delta^{18}O_{seawater}$ values of 0 ± 5‰ since 3.5 Ga. In this case, the oldest ocean would be much warmer, with temperatures 50–70 °C higher than today at 3.5 Ga[6,7]. Until now, the test for these two hypotheses (increasing or constant $\delta^{18}O_{seawater}$ over time) has hinged almost completely on the sedimentary isotope record. However, the sedimentary isotopic signal is known to be significantly affected by second-order processes such as evaporation, continental water percolation, and/or post-depositional interaction with pore water leading to important deviations from the true seawater value[6]. In the same way, the serpentinite isotopic composition may not reflect the seawater $\delta^{18}O$ too, because the provided signal is partially controlled by the silicate minerals during the serpentinization process[5,10]. Therefore, a more accurate determination of the bottom seawater $\delta^{18}O$ is still required to ultimately test the available hypotheses for $\delta^{18}O_{seawater}$ evolution, and consequently the ocean's temperature evolution.

Here we investigate the serpentinites of the ca. 760 Ma Bou Azzer ophiolite[11-16] (Anti-Atlas, Morocco), hosting massive and well-preserved magnetite veins[11,17,18]. We confirm that these veins derived from a black smoker-type abyssal hydrothermalism[18]. This high-quality isotopic archive consisting of pure magnetite from the massive veins allows us to propose an accurate $\delta^{18}O$ value for the Neoproterozoic bottom seawater.

## Results

**Relics of a black smoker-type hydrothermalism.** Serpentinites from the North Aït Ahmane unit of the Bou Azzer ophiolite[11-15] (ca. 760 Ma[15,16], Morocco) experienced an intense hydrothermal activity that produced unusually massive, up to 5 cm thick magnetite veins[11,17,18]. A detailed magneto-petrographic study[18] of the hydrothermalized serpentinites hosting the veins showed that an intense iron leaching in the serpentinites by a Cl-rich acidic fluid provided the iron for magnetite precipitation. Both abyssal and tardi-orogenic settings were proposed concerning the involved hydrothermal event[18]. Here we provide geochemical data on the serpentinites attesting that a black smoker-type (abyssal) hydrothermalism generated these unique magnetite veins.

Strong LREE and Eu enrichment are the hallmark of fluids exhaled by the present day black smoker-type abyssal hydrothermal vents[19,20] (Figs. 1 and 2). In ultramafic rocks, such REE patterns are reported only for serpentinites originated from such abyssal hydrothermal vent fields[21-24] (Figs. 1 and 2). Firstly, interpreted as the result of fluid/rock interaction with plagioclase-bearing mafic rocks[25-27], these LREE and Eu enrichments are now explained by the high mobility of these elements in acidic Cl-rich fluids, due to chlorine complexation at low pH[19,20]. Chlorine complexation is also advanced to explain the ability of such acidic Cl-rich fluids to mobilize and transport significant amounts of

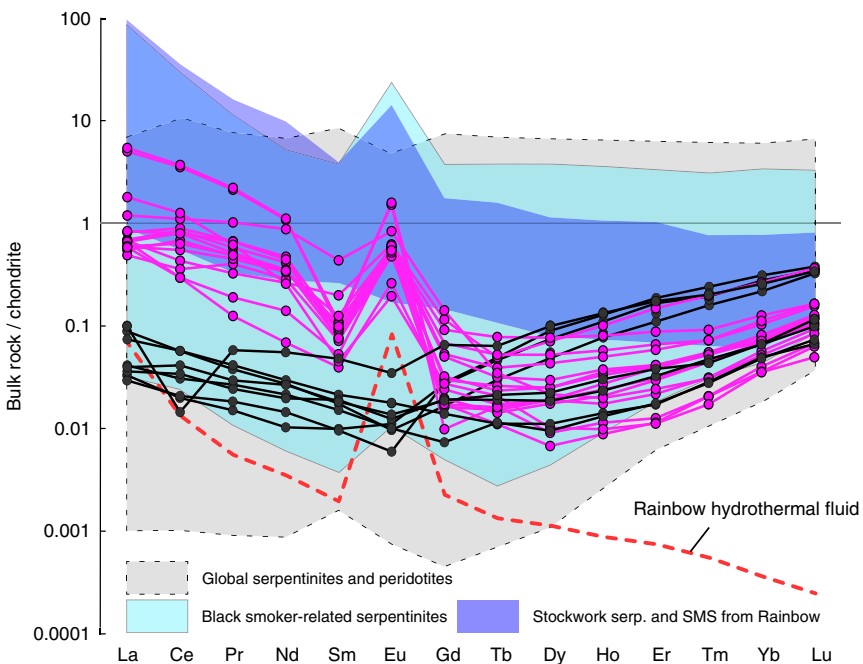

**Fig. 1** Chondrite-normalized REE compositions of North Aït Ahmane serpentinites. Hydrothermalized serpentinites (in pink) exhibit significant LREE enrichment and positive Eu anomaly contrasting with unaffected serpentinites (in black). Black smoker fluid of Rainbow (red dotted line) displays the same LREE and Eu pattern despite its lower REE content[19]. Serpentinites and peridotites from abyssal[21,28,61,62] and supra-subduction zone (SSZ)[62-65] settings, black smoker-related serpentinites[21-24], stockwork serpentinites and semi-massive sulphides from the Rainbow site[23] are shown for comparison. Chondrite normalization values are from Barrat et al.[66]

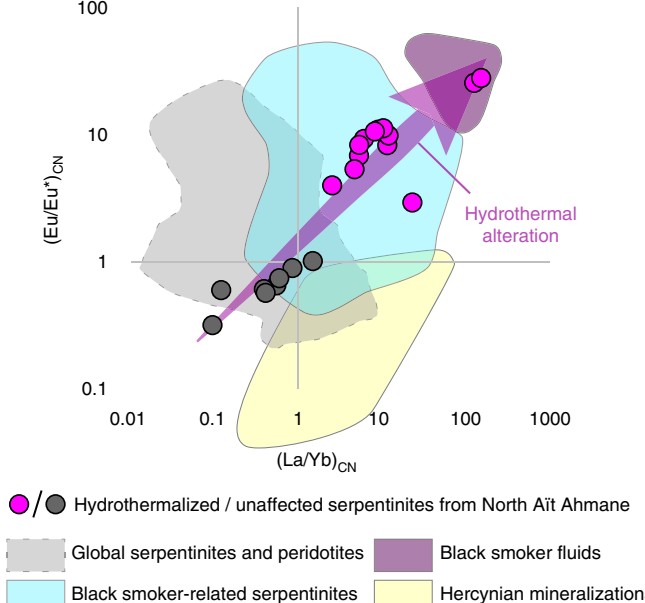

**Fig. 2** (Eu/Eu*)$_{CN}$ ratio vs. (La/Yb)$_{CN}$ ratio in North Aït Ahmane serpentinites. Data are compared with global serpentinites and peridotites[28,61,62,64,65] (in gray) and with black smoker-related serpentinites[21-24] (in blue). Correlated enrichment in LREE and Eu in hydrothermalized serpentinites from Aït Ahmane tends toward those of black smoker fluids sampled at the sites of Rainbow[19], Logatchev[19], and Manus basin[20] (in purple). Carbonates hosting tardi-orogenic mineralizations in Bou Azzer inlier, dated at 310 ± 5 Ma[30], plot in a clearly different field (in yellow), discarding a post-obduction process[18] for Eu and LREE enrichments in the serpentinites. Chondrite normalization values are from Barrat et al.[66]

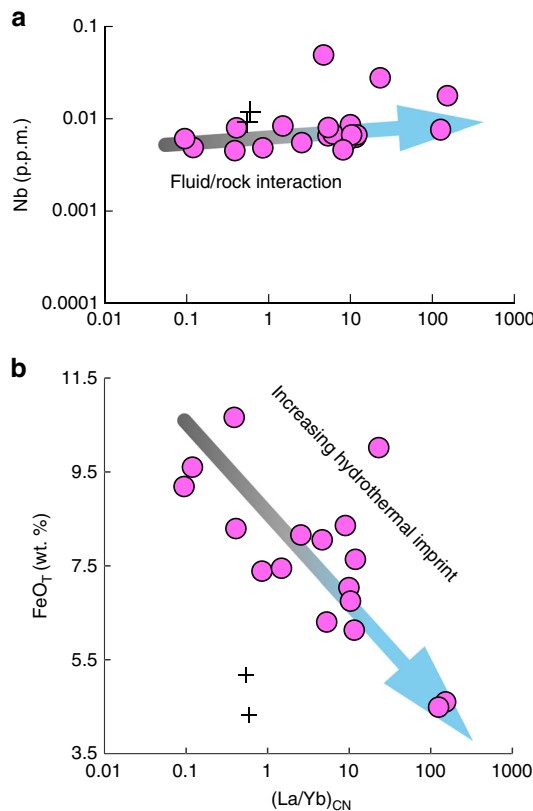

**Fig. 3** Nb and FeO$_T$ contents versus (La/Yb)$_{CN}$ ratio in North Aït Ahmane serpentinites. **a** Non-correlation between (La/Yb)$_{CN}$ ratio and Nb content attests to the hydrothermal origin of LREE enrichment. **b** Negative correlation between FeO$_T$ and (La/Yb)$_{CN}$ highlights the relationship between black smoker-type hydrothermalism and iron leaching in serpentinites involved in magnetite veins formation. Black crosses are antigorite veins. La and Yb are normalized to chondrite[66]

transition metals, including iron (forming Fe–Cl complexes), within abyssal hydrothermal systems[19]. Here, we report identical geochemical features for the hydrothermalized serpentinites of the North Aït Ahmane unit (Fig. 1; see Supplementary Data 1). These serpentinites clearly display strong LREE enrichments ([La/Yb]$_{CN}$ up to 152) correlated with positive Eu anomalies ([Eu/Eu*]$_{CN}$ up to 27.4), contrasting with unaffected serpentinites displaying classical U-shaped REE patterns (Figs. 1 and 2). Further support for the hydrothermal origin of this chemical signature, in opposition to magmatic refertilization processes, is provided by the lack of correlation between HFSE and LREE enrichments in our samples, since HFSE are immiscible in low-temperature aqueous solutions[21,28,29] (Fig. 3a). By contrast, the geochemical signature of carbonates related to the tardi-orogenic event[30] significantly differs (Fig. 2), ruling out a post-obduction setting for serpentinites' alteration and magnetite veins' genesis.

In addition, high As and Sb concentrations also characterize serpentinites from current black smoker vent fields due to hydrothermal fluid/rock interactions[23,24]. The high As and Sb contents of the North Aït Ahmane serpentinites (As: 0.43–224 p.p.m., Sb: 0.01–0.73 p.p.m.) are akin to As and Sb contents in these modern black smoker hosted serpentinites[23,24]. A late sedimentary origin for these high As concentrations is very unlikely, given the absence of correlated LILE enrichments[31].

Interestingly, accessory minerals are also affected in current black smoker systems, such as Cr-spinels, which are extensively altered and display important Mn-rich ferritchromite alteration rims (up to 4.53 wt.% MnO)[32]. As previously shown by Hodel et al.[18], the hydrothermal alteration of the North Aït Ahmane serpentinites also drastically affected the Cr-spinels they host. Ferritchromite and Cr-magnetite rims resulting from this

alteration are highly enriched in Mn, up to 5.41 wt.% of MnO[18,33], which is once again exclusive to black smoker-related serpentinites[32]. Finally, samples presenting the highest LREE enrichments and the strongest Eu anomalies are characterized by a high abundance of sulphides and can be analogous to the sulfide-rich serpentinites and stockworks of modern black smoker systems[23,24].

In sum, all these petrographical and geochemical features indicate that North Aït Ahmane serpentinites endured an abyssal black smoker-type hydrothermalism before the obduction of the ophiolitic sequence. Magnetite veins' formation from iron leaching by acidic Cl-rich fluid[18] in these serpentinites clearly results from this abyssal hydrothermalism, as further evidenced by a negative correlation between total iron content and (La/Yb)$_{CN}$ ratio (Fig. 3b). Thus, these massive magnetite veins and the associated hydrothermalized serpentinites likely represent the oldest fossil ultramafic-hosted black smoker-type hydrothermal system ever described.

**Temperature of the involved fluid.** The absence of antigorite in the studied hydrothermalized serpentinites constrains the temperature of the involved fluid to below 350 °C[34,35]. Clinochlore blades resulting from Cr-spinel alteration during the abyssal hydrothermal event[18] can be used to precisely infer its temperature using chlorite thermometry[36–39]. Here, we used the semi-empirical chlorite thermometer of Lanari et al.[36], which is based on a recent thermodynamic model for di-trioctahedral chlorite from experimental and natural data in the system MgO-

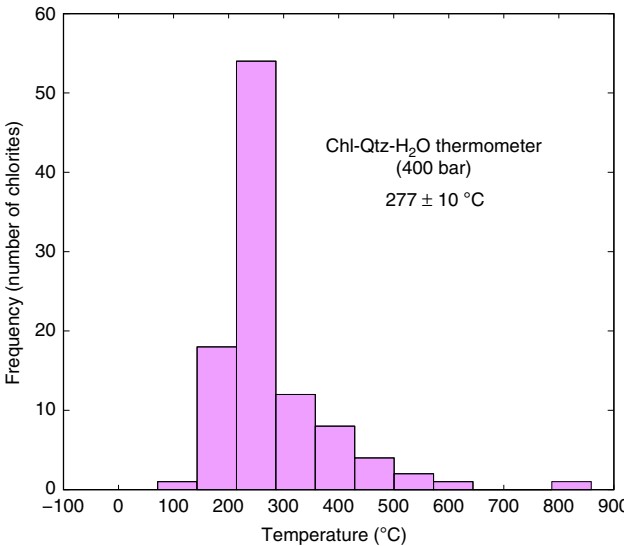

**Fig. 4** Calculated temperature for the hydrothermal event. Bootstrap statistical analysis[40] on chlorite temperatures calculated at 400 bar using the Chlorite-Qtz-$H_2O$ thermometer[36]

$FeO-Al_2O_3-SiO_2-H_2O$[36–38]. Temperatures were calculated for a range of four different pressures (between 300 bar and 2.5 kbar) using bootstrap statistical analysis to set aside badly crystallized chlorites[40] (Fig. 4; see Supplementary Data 2). Since black smoker-type hydrothermalism occurs at the seafloor subsurface, the most reasonable associated pressure must be 400 bar, which corresponds to a 3000 m water column and a 1 km fluid penetration depth within the oceanic lithosphere[24]. It is worth noting that chlorite crystallization due to Cr-Spinel hydrothermal alteration is closely linked with magnetite veins' precipitation, the involved hydrothermal fluid in both cases being likely the same[18]. Thus, the temperature of $277 \pm 10\,°C$ assessed by chlorite thermometry is also the temperature of the hydrothermal fluid during magnetite veins' precipitation and can be used to constrain the oxygen isotopes fractionation between hydrothermal fluid and magnetite.

**$\delta^{18}O$ of Neoproterozoic seawater.** In the North Aït Ahmane black smoker system, iron was leached out from the host serpentinites by the hydrothermal fluid and transported as Fe–Cl complexes up to the cracks, where it precipitated in the form of iron oxide[18–20,41–43]. Hence, the large amount of oxygen required to precipitate magnetite ($Fe_3O_4$) as massive veins directly stems from the seawater-derived hydrothermal fluid circulating in the system. Consequently, the magnetite isotopic oxygen composition is necessarily in equilibrium with this hydrothermal fluid. Therefore, massive pure magnetite veins formed during this abyssal hydrothermalism are ideal targets to assess the $\delta^{18}O$ of Neoproterozoic seawater. We measured the oxygen isotopic composition of pure magnetite ($\delta^{18}O_{Mgt}$) from five magnetite veins. $\delta^{18}O_{Mgt}$ values range from $-9.33$ to $-8.16‰$ with a mean value of $-8.95 \pm 0.42‰$ (Fig. 5; see Supplementary Table 1). The mean $\delta^{18}O$ for the fluid ($\delta^{18}O_{fluid}$) in equilibrium with the magnetite of these veins was calculated at $-0.42 \pm 0.55‰$ using a Mgt-$H_2O$ fractionation law[44] and the temperature set at $277 \pm 10\,°C$ (Fig. 5; see Methods, Supplementary Table 1 and Supplementary Fig. 1). For comparison, a maximum temperature of $302 \pm 11\,°C$ (at 2.5 kbar) and a minimum temperature of $273 \pm 10\,°C$ (at 300 bar) would provide similar $\delta^{18}O_{fluid}$ values, respectively $-0.52 \pm 0.54‰$ and $-0.41 \pm 0.56‰$, attesting the robustness of our estimate.

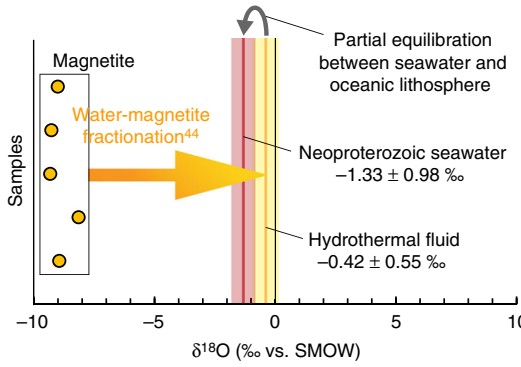

**Fig. 5** Oxygen isotope compositions of the hydrothermal magnetite. Measured $\delta^{18}O$ compositions of pure magnetite from five veins (orange dots), assessment of the isotopic composition of the involved hydrothermal fluid and calculation of the corresponding seawater isotopic composition

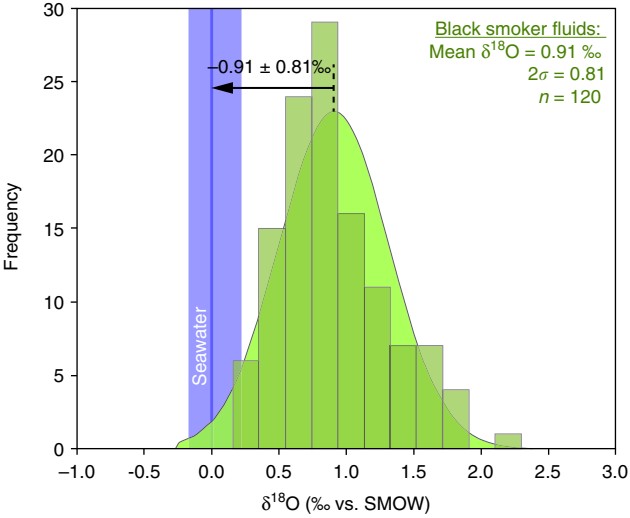

**Fig. 6** Partial $\delta^{18}O$ equilibration between present-day bottom seawater and black smoker fluids. Distribution histogram and associated curve (in green) for a compilation of black smoker fluids' $\delta^{18}O$ values ($n = 120$)[45–48]. Present-day bottom seawater range is represented in blue[46,47]

The value of $-0.42 \pm 0.55‰$ we obtained for the hydrothermal fluid in equilibrium with the magnetite does not directly correspond to the $\delta^{18}O$ of the Neoproterozoic bottom seawater. It has been shown that high-temperature abyssal hydrothermal fluids are enriched in $^{18}O$ relative to bottom seawater due to partial isotopic equilibration with mafic/ultramafic rocks of the oceanic lithosphere[45]. In order to quantify this partial equilibration, we compiled $\delta^{18}O$ data from present-day black smoker fluids[45–48] and associated bottom seawater values, respectively, $\delta^{18}O_{BSfluid}$ and $\delta^{18}O_{PDseawater}$ (Fig. 6). $\delta^{18}O_{PDseawater}$ values[46,47] are comprised between $-0.17$ and $0.22‰$, while $\delta^{18}O_{BSfluid}$ values ($n = 120$) range from $0.16$ to $2.30‰$. These data attest to a high degree of overlap between $\delta^{18}O_{PDseawater}$ and $\delta^{18}O_{BSfluid}$ with a small difference between their mean values (Fig. 6). Because of the normal statistical distribution of $\delta^{18}O_{BSfluid}$ data (Fig. 6), the mean shift between $\delta^{18}O_{BSfluid}$ and $\delta^{18}O_{PDseawater}$ can be used to quantify the fluid/rock equilibration. We obtained a $\Delta^{18}O_{BSfluid-PDseawater}$ of $0.91 \pm 0.81‰$ (Fig. 6).

Hence, a $\delta^{18}O$ value for the Neoproterozoic bottom seawater can be estimated from our isotopic measurements on the North Aït Ahmane magnetite veins by subtracting this $\Delta^{18}O_{BSfluid-}$

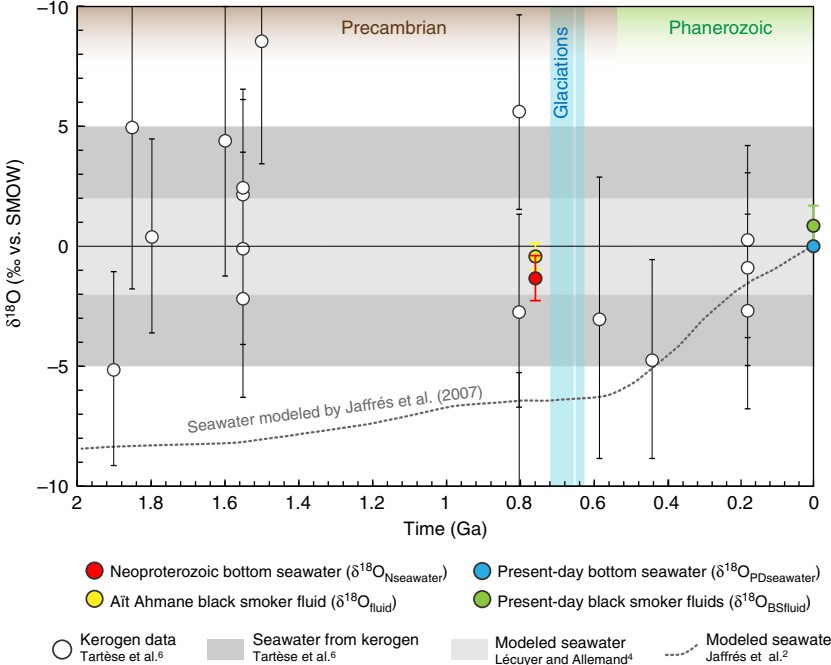

**Fig. 7** Oxygen isotopic compositions of the Aït Ahmane hydrothermal fluid and estimate of Neoproterozoic bottom seawater $\delta^{18}O$ compared with existing estimates along the past 2 Ga. Error bars for $\delta^{18}O$ values of Aït Ahmane black smoker fluid and Neoproterozoic seawater were calculated following error propagation considering errors on mean $\delta^{18}O$ value measured on magnetite, on temperature-dependent water–magnetite fractionation and on $\Delta^{18}O_{BSfluid-PDseawater}$ ($2\sigma$ of the compiled data, Fig. 6), see Supplementary Table 1

PDseawater value to the $\delta^{18}O_{fluid}$ calculated from magnetite veins (Fig. 5). In this manner, we obtained a $\delta^{18}O_{Nseawater}$ value of $-1.33 \pm 0.98$‰ for the Neoproterozoic bottom seawater at 760 Ma (Figs. 5 and 7; see Supplementary Table 1).

## Discussion

Global seawater $\delta^{18}O$ is essentially controlled by seafloor hydrothermal alteration, meaning the interaction of seawater with oceanic lithosphere in hydrothermal systems[2–4,8,9]. The ratio between high and low-temperature alteration has been evoked as maintaining the seawater $\delta^{18}O$ constant through time[4,8,9], partial isotopic equilibration mentioned above (Figs. 5 and 6) acting as a buffer. Some authors, however, argued that this ratio evolved through time due to a two-step rise of high-temperature abyssal hydrothermalism related to geodynamic changes since the Archean[2]. Long-term changes in sea level are also evoked as having interfered in the $\delta^{18}O$ regulation by changing the continental surface exposed to weathering[2]. Changes in sea level also influence water pressure at the bottom of the sea, regulating the depth of fluids penetration in mid-ocean ridge hydrothermal systems and thus the extent of high-temperature alteration[2,3]. The Neoproterozoic $\delta^{18}O_{seawater}$ of $-1.33 \pm 0.98$‰ that we provide here is much more precise than previous estimates (Fig. 7). This $\delta^{18}O_{Nseawater}$ value is significantly higher than that of $-6.4$‰ predicted by models considering an increasing $\delta^{18}O_{seawater}$ through time[2]. It means that the isotopic oxygen compositions of the Neoproterozoic oceans (at 760 Ma) was similar to that of the recent oceans, which is between $-1.5$ and $+1.8$‰[49], and $-1.4$‰ for an ice-free planet[50]. This result attests to a rather constant $\delta^{18}O_{seawater}$ through time, at least since the Neoproterozoic. Hence, the ratio of low- and high-temperature hydrothermal activity, ocean volume, ridge depth, and global geodynamics would have been similar than in the present day, meaning that a modern tectonic–ocean system already prevailed at 760 Ma.

Finally, the $\delta^{18}O_{seawater}$ presented here can be used to better constrain the temperature of the oceans at 760 Ma when combined with the available oxygen isotope record of authigenic carbonates and cherts. Past ocean temperatures can be estimated from the isotopic fractionation between seawater and marine sediments. These sedimentary records display a general trend of increasing $\delta^{18}O$ values from the Archean to the present[2,4,51–54]. Considering a steady $\delta^{18}O_{seawater}$ since the Archean, this $\delta^{18}O$ increase of carbonates and cherts is generally interpreted as resulting from the progressive cooling of the ancient oceans (from ~70 to 50 °C during the Archean)[4,6]. By evoking the implausibility of such high temperatures, some authors proposed that the isotopic signal of these sedimentary archives could have been modified by pervasive alteration processes such as diagenesis, post-depositional interaction with pore water or hydrothermal fluids on the seafloor[10,55]. Nonetheless, the fact that this trend is recorded in different mineralogies (carbonate, dolostone, chert, and phosphates)[2,3,6,7] and in different isotope systems (e.g., $\delta^{30}Si$[7], $\delta^{18}O$[2,6]) attests to the reliability of these isotopic records. Given this, our validation of a constant $\delta^{18}O_{seawater}$ indicates that the oceans were likely 15–30 °C warmer than today 760 Myr ago, on the eve of the events of life diversification that occurred at the end of the Neoproterozoic.

## Methods

**Chlorite thermometry.** Major element compositions of clinochlore associated with hydrothermal alteration[18] were determined with a Cameca SXFive electron microprobe at the Centre de Micro Caractérisation Raimond Castaing (Université Toulouse III Paul Sabatier, France). Operating conditions were as follows: accelerating voltage 15 kV and beam current 10 nA. Analyzed surface is ~2 × 2 μm². The following standards were used: albite (Na), periclase (Mg), corundum (Al), sanidine (K), wollastonite (Ca, Si), pyrophanite (Mn, Ti), hematite (Fe), $Cr_2O_3$ (Cr), NiO (Ni), sphalerite (Zn), tugtupite (Cl), barite (Ba), and topaze (F). Detection limits are estimated to be 0.01 wt.% for each element. Temperatures were assessed using the semi-empirical geothermometer of Lanari et al.[36] (see Supplementary Data 2). Temperatures were calculated for a range of four different pressures (300, 400 bar, 1, 2.5 kbar) to test the quality of the thermometer and gives very similar

results: $273 \pm 10$ °C at 300 bar, $277 \pm 10$ °C at 400 bar, $283 \pm 10$ °C at 1 kbar, and $302 \pm 11$ °C at 2.5 kbar.

**$\delta^{18}$O measurements and calculation.** Oxygen in pure magnetite from five massive magnetite veins was extracted as $O_2$ gas via reaction with bromine pentafluoride ($BrF_5$). Oxygen ratios were measured on a Thermo Fisher Scientific Delta Plus XP mass spectrometer at the Institut de Physique du Globe de Paris (Paris, France). All values of $\delta^{18}$O $[ = [(^{18}O/^{16}O)_{sample}/(^{18}O/^{16}O)_{standard} - 1] \times 1000]$ are normalized to VSMOW. NBS 28 reference material was measured at $9.48 \pm 0.10$‰ ($\sigma$, $n = 7$) during the course of this study. We tested two Mgt–$H_2O$ fractionation laws[44,56] to assess the $\delta^{18}$O of the hydrothermal fluid involved in magnetite vein precipitation at $277 \pm 10$ °C. In order to validate the predictions of these laws, we used the direct measurements of Fortier et al.[57] that provide a value of reference at 350 °C. Both laws predict a very similar Mgt–$H_2O$ fractionation consistent with Fortier et al.[57] measurements (see Supplementary Fig. 1). We retained the fractionation law proposed by Zheng and Simon[44] because it is better suited for temperature estimated for magnetite veins precipitation ($277 \pm 10$ °C). Law by Cole et al.[56] is limited to 300 °C and has large uncertainties below 500 °C.

**Bulk-rock chemistry.** Major element concentrations were obtained by alkaline fusion and ICP-OES analysis at the Service d'Analyse de Roches et des Minéraux (Nancy, France) following the analytical protocol of Carignan et al[58]. Whole-rock trace element concentrations were analyzed following the procedure of Ionov et al.[59] adapted by for the analysis of ultra-depleted peridotites[60] using an Agilent 7700x quadrupole ICP-MS at Géosciences Montpellier (Université Montpellier 2, France). Indium and Bismuth were used as internal standards during ICP-MS measurements. The precision and accuracy of ICP-MS analyses were assessed by measuring the reference materials BE-N (basalt) and UB-N (serpentinite).

**Data availability.** All data used in this manuscript are available in Supplementary Data 1, Supplementary Data 2, and Supplementary Table 1. Further queries and information requests should be directed to the lead author F.H. (florent.hodel@hotmail.fr).

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

## Acknowledgements

We thank Olivier Bruguier, Chantal Douchet, and Léa Causse for their assistance on the ICP-MS. This work has been funded by TellUS-SYSTER program of INSU (CNRS), French MENESR, and Research Grant 2016/06114-6 of the São Paulo Research Foundation (FAPESP).

## Author contributions

F.H., M.M., and R.I.F.T. conceived the study and wrote the paper with contributions from all co-authors. Field work and sampling were done by F.H., A.T., M.M., J.B., and N. E.; trace elements' chemistry was performed by F.H. and J.B. and interpreted by F.H., M. R., V.C., and J.C.; J.G. and F.H. did the thermometry study on chlorites. P.A. and F.H. performed the oxygen isotope measurements and C.D. did the statistics and error calculations.

## Additional information

**Competing interests:** The authors declare no competing interests.

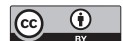

