## [Peer Review File · Nature Communications]

Reviewers' comments:

Reviewer #1 (Remarks to the Author):

The authors of this paper use oxygen isotope measurements of magnetite veins in 760-700 Ma ophiolites to argue that the isotopic composition of the Neoproterozoic oceans was close to modern. Specifically, the magnetite itself has $\delta^{18}\text{O} = 8.95 \pm 0.42\text{‰}$, while seawater is predicted to have $\delta^{18}\text{O} = 0.42 \pm 0.55\text{‰}$.

The first question, of course, is whether the studied section is really an ophiolite. I am not expert in these determinations, so I will take the authors' word for it. The next question is whether their assumption that thermodynamic equilibrium would have prevailed between the magnetite vein and seawater. The authors assume that it did, then they use a corresponding thermodynamic equilibrium fractionation formula to estimate the difference between seawater $\delta^{18}\text{O}$ and the $\delta^{18}\text{O}$ of the magnetite. The temperature of the interaction is determined to be $277 \pm 10\text{°C}$, based on a new chlorite thermometry proxy developed by the first author in an earlier paper. This, I think, is a fundamental flaw with the paper. What evidence do the authors have that thermodynamic equilibrium would really be achieved between the magnetite and the seawater flowing through the hydrothermal system? The authors talk about the enormous amount of O needed to oxidize iron and make magnetite. But $\frac{3}{4}$ of the O in magnetite, Fe_3O_4 , comes from the original FeO in the basalt, while $\frac{1}{4}$ of it comes from seawater. Consider what would happen if thermodynamic equilibrium was not achieved. The oxygen in the original FeO should have $\delta^{18}\text{O} = +5.7\text{‰}$ (the mantle value). To achieve $\delta^{18}\text{O} = 8.95\text{‰}$ for the bulk magnetite, the oxygen added from seawater would have to have $\delta^{18}\text{O} = -26\text{‰}$, by mass balance. Now, this estimate is admittedly extreme because the authors are probably at least partially correct, i.e., there should be some back-and-forth exchange of oxygen between the magnetite and the seawater flowing through the vents. But if equilibrium is not achieved, then the seawater could be isotopically much lighter than reported here.

Full disclosure from this reviewer: We have a paper that is struggling through the review process in which we argue that isotopic equilibrium is not achieved between seawater and the rocks through which it circulates within the midocean spreading ridges. The water in the pore spaces of the rock does not exchange that rapidly with the water flowing through the vents, and the rock itself is always moving away from the ridge axis. Thus, it spends a limited amount of time exchanging oxygen with the ventwater.

Our paper cannot be used as a strong argument, obviously, because it is not published. But the authors, and the editor, should carefully consider whether thermodynamic equilibrium really prevails in this system. After all, the bottom line of this paper is that ocean temperatures were 15-30°C warmer than today between 760-700 Ma. Meanwhile, the Sturtian glaciation, which was a snowball event according to most researchers, is dated at 720-660 Ma, again according to the present authors. If their paper is correct, then the dates have to be adjusted so that the ophiolite studied here occurs prior to the glaciation. And someone needs to explain why the Earth cooled dramatically within a geologically short time interval to set up the glaciation. It seems much more plausible to this reviewer that the oceans were already cool by the late Proterozoic and that the problem lies in the interpretation of the oxygen isotope record, which could be incorrect for the reasons given above.

You may reveal my name to the authors.

Jim Kasting

Reviewer #2 (Remarks to the Author):

Comments on "Fossil black smoker yields oxygen isotopic composition of Neoproterozoic seawater" by Hodel et al.

This ms reports $\delta^{18}\text{O}$ analyses of 5 samples of magnetite veins from the 760-700 Ma old Bou Azer

rocks in Morocco. The ophiolitic nature of the Bou Azer rocks is indicated by their LREE and trace elements signatures. The $\delta^{18}\text{O}$ of the magnetite veins are homogeneous and are used to back calculate the $\delta^{18}\text{O}$ of the ambient hydrothermal fluid at the time of their precipitation (assuming a given magnetite-fluid oxygen isotopic fractionation law, and a given T of formation), and in a second step, the $\delta^{18}\text{O}$ of ambient seawater (assuming a given $\delta^{18}\text{O}$ difference between average hydrothermal fluids in black smokers and seawater).

The results show a $\delta^{18}\text{O}$ of seawater similar than today and bring support to previous studies and theoretical considerations showing that the $\delta^{18}\text{O}$ of seawater has not varied by more than 1-2 permil over geological times.

The topic of this paper is interesting. The question of the surface temperature of the Earth in the Precambrian is very important. This is related to the question of the oxygen isotopic composition of seawater at that time. There are few cases where the $\delta^{18}\text{O}$ of past seawater can be constrained. Thus this ms could be very well suited for publication in Nature geosciences.

However, I see a major problem in this study, which prevents its publication in Nature geosciences or in any journal, until this problem is solved. The magnetite veins are presented as if they were precipitated in a black smoker environment and would thus give access to coeval seawater. But there is NO decisive argument given in this ms for that. There is NO decisive proof that the $\delta^{18}\text{O}$ which is reconstructed is for seawater and not for a continental hydrothermal fluid. This seems to me to be a critical problem.

In addition, I find it extremely embarrassing that the first author of this ms has just published in Precambrian Research in 2017 (Hodel et al. Precambrian Research 2017, 300, 151-167) a paper saying the contrary of what is said in this ms.

The abstract of this paper clearly says that there are two possibilities for the origin of the magnetite veins: " (1) a continental hydrothermal system as advanced for the Co-Ni-As ores in the Bou Azer inlier or (2) an oceanic black smoker type hydrothermal vent field on the Neoproterozoic seafloor. "

In this paper, section 5.4 clearly describes and gives several convincing arguments for the first possibility to be correct (the same is done for the second, showing that no definitive conclusion can be reached):

" (1) Several authors (Bouabdellah et al., 2016 and references therein) consider the Co-Ni-Fe-As-(Ag-Au) ores in the Bou Azer inlier as a product of a polyphased hydrothermal system affecting the ultramafics between 380 and 240 Ma (Gasquet et al., 2005). These mineralizations (forming quartz-carbonates hosted ores) are typically localized at the contact between the serpentinites and the late quartz-diorite that intruded the massif (650–640 Ma, Inglis et al., 2005; Fig. 1. b). More precisely, these authors attribute the Co-Ni-Fe-arsenide mineralizations, to an intermediate phase involving Ca, Cl-rich (36–45 wt% NaCl + CaCl equiv., Bouabdellah et al., 2016) magmatic/hydrothermal fluids, potentially mixed with meteoric water at $T < 200$ °C. There is nowadays a consensus on the fact that serpentinites are the sources for Co-Ni and Fe (e.g. Ahmed et al. 2009b; Bouabdellah et al., 2016). The high chlorinity reported for the mineralizing fluids suggests a chloride complexation to explain the metals transport (e.g. Bouabdellah et al., 2016). An interaction with such fluid is denoted by the relative high chlorine concentrations in the Ait Ahmane hydrothermalized serpentinites hosting the massive magnetite veins by comparison to the unaltered ones (Figs. 6 and 13). These fluids could have mobilized the transition metals (Co, Ni and particularly Fe) in the serpentinites and precipitate Co, Ni-rich magnetite veins in cracks due to pressure drop or fluid mixing prompting the precipitation."

In the present ms, the arguments given are not decisive to my opinion (and there is no mention made of the other hypothesis described in Hodel et al 2017). Figs 1 and 2 and the major part of the discussion on this subject relates to the serpentinites, but I think that this is not the critical point: the serpentinites have been hydrothermalized in an oceanic setting, the question is the origin of the magnetite veins. As discussed above, this can take place in a very different setting, a few

100 Ma later. Fig 3 brings argument to make the point of the authors but it is not very strong to me.

Otherwise all the isotopic discussion is correct, and the reconstructed $\delta^{18}\text{O}$ for the hydrothermal fluid is correct.

In conclusion to make it clear, it seems to me that:

- there is for sure an ophiolite at 760-700 Ma (even if it is not said how it is precisely dated)
- there is for sure oceanic serpentinization with hydrothermal seawater derived fluids that caused formation of a first generation of magnetites in the serpentinites
- then there was a hydrothermal event that remobilized Fe and produced the magnetite veins.

There is no decisive proof in the ms that this event was in an oceanic setting. There seems to be no proof either that it was at 700 Ma, but could have been at 380-240 Ma.

Detailed response to reviewers' comments:

Jim Kasting

Reviewer #1 (Remarks to the Author):

“The authors of this paper use oxygen isotope measurements of magnetite veins in 760-700 Ma ophiolites to argue that the isotopic composition of the Neoproterozoic oceans was close to modern. Specifically, the magnetite itself has $\delta^{18}\text{O} = 8.95 \pm 0.42\text{‰}$, while seawater is predicted to have $\delta^{18}\text{O} = 0.42 \pm 0.55\text{‰}$.”

“The first question, of course, is whether the studied section is really an ophiolite. I am not expert in these determinations, so I will take the authors' word for it.”

R1. The Bou Azzer ophiolitic complexes are well known in the literature (e.g. Leblanc, 1975; Bodinier et al., 1984; Naidoo et al., 1991; Walsh et al., 2012; Hodel et al., 2017; Hodel, 2017; Triantafyllou et al., 2018) and had been included in almost all compilations (e.g. Furnes et al., 2014, 2015). Therefore, there is no doubt it is an ophiolite.

L. 42, we added some key references in order to eliminate any doubt about the ophiolitic nature of the studied unit: Bodinier et al., 1984; Naidoo et al., 1991; Walsh et al., 2012; Hodel et al., 2017; Hodel, 2017; Triantafyllou et al., 2018.

“The next question is whether their assumption that thermodynamic equilibrium would have prevailed between the magnetite vein and seawater. The authors assume that it did, then they use a corresponding thermodynamic equilibrium fractionation formula to estimate the difference between seawater $\delta^{18}\text{O}$ and the $\delta^{18}\text{O}$ of the magnetite.”

R2. This point is detailed below in R4.

“The temperature of the interaction is determined to be $277\pm 10^\circ\text{C}$, based on a new chlorite thermometry proxy developed by the first author in an earlier paper. This, I think, is a fundamental flaw with the paper.”

R3. This geothermometer was conceived by Lanari et al (2014) and was successfully used in previous papers (e.g. Block et al., 2015). It was not developed by F. Hodel but by an independent group (Olivier Vidal and collaborators) and F. Hodel et al. didn't use chlorite thermometry in their previous paper.

We rewrote the following part of the main text to clarify this misunderstanding:

L. 126-131: *“Clinochlore blades resulting from Cr-spinel alteration during the abyssal hydrothermal event¹⁸ can be used to precisely infer its temperature using chlorite thermometry³⁶⁻³⁹. Here, we used the semi-empirical chlorite thermometer of Lanari et al.³⁶, which is based on a recent thermodynamic model for ditrioctahedral chlorite from experimental and natural data in the system MgO-FeO-Al₂O₃-SiO₂-H₂O³⁶⁻³⁸.”*

“What evidence do the authors have that thermodynamic equilibrium would really be achieved between the magnetite and the seawater flowing through the hydrothermal system? The authors talk about the enormous amount of O needed to oxidize iron and make magnetite. But $\frac{3}{4}$ of the O in magnetite, Fe₃O₄, comes from the original FeO in the basalt, while $\frac{1}{4}$ of it comes from seawater. Consider what would happen if thermodynamic equilibrium was not achieved. The oxygen in the original FeO should have $\delta^{18}\text{O} = +5.7\text{‰}$ (the mantle value). To achieve $\delta^{18}\text{O} = 8.95\text{‰}$ for the bulk magnetite, the oxygen added from seawater would have to have $\delta^{18}\text{O} = \sim -26\text{‰}$, by mass balance. Now, this estimate is admittedly extreme because the authors are probably at least partially correct, i.e., there should be some back-and-forth exchange of oxygen between the magnetite and the seawater flowing through the vents. But if equilibrium is not achieved, then the seawater could be isotopically much lighter than reported here.”

Full disclosure from the reviewer: “We have a paper that is struggling through the review process in which we argue that isotopic equilibrium is not achieved between seawater and the rocks through which it circulates within the midocean spreading ridges. The water in the pore spaces of the rock does not exchange that rapidly with the water flowing through the vents, and the rock itself is always moving away from the ridge axis. Thus, it spends a limited amount of time exchanging oxygen with the ventwater. Our paper cannot be used as a strong argument, obviously, because it is not published. But the authors, and the editor, should carefully consider whether thermodynamic equilibrium really prevails in this system.”

R4. The main point raised by the reviewer is if thermodynamic equilibrium would really be achieved between the magnetite and the seawater-derived fluid flowing through the hydrothermal system. The reviewer says: *“The authors talk about the enormous amount of O needed to oxidize iron and make magnetite. But $\frac{3}{4}$ of the O in magnetite, Fe₃O₄, comes from the original FeO in the basalt, while $\frac{1}{4}$ of it comes from seawater. Consider what would happen if thermodynamic equilibrium was not achieved. The oxygen in the original FeO should have $\delta^{18}\text{O} = +5.7\text{‰}$ (the mantle value). To achieve $\delta^{18}\text{O} = 8.95\text{‰}$ for the bulk magnetite, the oxygen added from seawater would have to have $\delta^{18}\text{O} = \sim -26\text{‰}$, by mass balance.”* Contrary to this affirmation, FeO is known to be immobile in aqueous systems. This is a consensus in the specialized literature (e.g. Purtoev et al., 1989; Fein et al., 1992; Douville et al., 2002; Craddock et al., 2010; Kalczynski et al., 2014). The only way to transport iron in aqueous

solutions is by complexation with strong anionic ligands such as chlorine or fluorine. So, in the case studied here, iron reaches the cracks in the serpentinites in the form of FeCl_2^{2+} , FeCl_2^+ or FeCl_3 and the oxygen incorporated during magnetite precipitation must come from the aqueous hydrothermal fluid. As a consequence, the magnetite isotopic oxygen composition is obviously in equilibrium with the hydrothermal fluid.

We added the following sentences in the main text to clarify this point:

L. 63-66: “Chlorine complexation is also advanced to explain the ability of such acidic Cl-rich fluids to mobilize and transport significant amounts of transition metals, including iron (forming Fe-Cl complexes), within abyssal hydrothermal systems¹⁹.”

L. 142-148: “In the North Aït Ahmane black smoker system, iron was leached out from the host serpentinites by the hydrothermal fluid and transported as Fe-Cl complexes up to the cracks, where it precipitated in the form of iron oxyde^{18-20,41-43}. Hence, the large amount of oxygen required to precipitate magnetite (Fe_3O_4) as massive veins stems directly from the seawater-derived hydrothermal fluid circulating in the system. Consequently, the magnetite isotopic oxygen composition is necessarily in equilibrium with this hydrothermal fluid.”

We agree with the affirmation (full disclosure of an unpublished paper) that “isotopic equilibrium is not achieved between seawater and the rocks through which it circulates within the midocean spreading ridges”. We explicitly consider this effect of partial equilibration when we recalculated the seawater $\delta^{18}\text{O}$ from the hydrothermal fluid. For that we assessed the isotopic compositional shift between hydrothermal fluids and seawater from a thorough compilation of 120 present-day black smokers and the corresponding seawater composition (Fig. 6).

In order to better explain our approach, we rewrote the following parts of the main text and added two figures (Fig. 5 and Fig. 6, the last one being initially in the supplementary materials):

L. 157-168: “The value of -0.42 ± 0.55 ‰ we obtained for the hydrothermal fluid in equilibrium with the magnetite does not directly correspond to the $\delta^{18}\text{O}$ of the Neoproterozoic bottom seawater. It has been shown that high temperature abyssal hydrothermal fluids are enriched in ^{18}O relative to bottom seawater due to partial isotopic equilibration with mafic/ultramafic rocks of the oceanic lithosphere⁴⁵. In order to quantify this partial equilibration, we compiled $\delta^{18}\text{O}$ data from present-day black smoker fluids⁴⁵⁻⁴⁸ and associated bottom seawater values, respectively $\delta^{18}\text{O}_{\text{BSfluid}}$ and $\delta^{18}\text{O}_{\text{PDseawater}}$ (Fig 6). Present-day $\delta^{18}\text{O}_{\text{PDseawater}}$ values^{46,47} are comprised between -0.17 and 0.22 ‰ while the $\delta^{18}\text{O}_{\text{BSfluid}}$ values ($n=120$) range from 0.16 to 2.30 ‰. These data attest to a high degree of overlap between $\delta^{18}\text{O}_{\text{PDseawater}}$ and $\delta^{18}\text{O}_{\text{BSfluid}}$ with a small difference between their mean values (Fig. 6). Because of the normal statistical distribution of $\delta^{18}\text{O}_{\text{BSfluid}}$ data (Fig. 6), the mean shift between $\delta^{18}\text{O}_{\text{BSfluid}}$ and $\delta^{18}\text{O}_{\text{PDseawater}}$ can be used to quantify the fluid/rock equilibration. We obtained a $\Delta^{18}\text{O}_{\text{BSfluid-PDseawater}}$ of 0.91 ± 0.81 ‰ (Fig. 6).

L. 172-176: “Hence, a $\delta^{18}\text{O}$ value for the Neoproterozoic bottom seawater can be estimated from our isotopic measurements on the North Aït Ahmane magnetite veins by subtracting this $\Delta^{18}\text{O}_{\text{BSfluid-PDseawater}}$ value to the $\delta^{18}\text{O}_{\text{fluid}}$ calculated from

magnetite veins (Fig. 5). In this manner, we obtained a $\delta^{18}\text{O}_{\text{Nseawater}}$ value of $-1.33 \pm 0.98 \text{ ‰}$ for the Neoproterozoic bottom seawater at 760 Ma (Figures 5 and 7)."

“After all, the bottom line of this paper is that ocean temperatures were 15-30°C warmer than today between 760-700 Ma. Meanwhile, the Sturtian glaciation, which was a snowball event according to most researchers, is dated at 720-660 Ma, again according to the present authors. If their paper is correct, then the dates have to be adjusted so that the ophiolite studied here occurs prior to the glaciation. And someone needs to explain why the Earth cooled dramatically within a geologically short time interval to set up the glaciation. It seems much more plausible to this reviewer that the oceans were already cool by the late Proterozoic and that the problem lies in the interpretation of the oxygen isotope record, which could be incorrect for the reasons given above.”

R5. The bottom line of the paper is to provide an unprecedentedly precise oxygen isotopes estimate of the Neoproterozoic seawater. High quality data would be the base to draw conclusions on the evolution of the seawater temperature. So, if we combine our results with the carbonates and cherts oxygen isotope database like previous authors have done (e.g. Jaffrés et al., 2007; Tartèse et al., 2017), one of the consequences of our result is that ocean temperatures were higher than today. It is true that such a conclusion requires an assessment of the reliability of the oxygen isotopic composition of ancient carbonates, and this is out the scope of our paper.

We clarified our discussion in this revised version by rewriting the last part of the paper but it does not reduce the strength of our message. In addition, recent works of Hodel (2017) and Triantafyllou et al. (2016, 2018) also allowed to show a clear link between the Aït Ahmane ophiolite (studied here) and the Khzama ophiolite (in the Sirwa inlier ~100 km to the NW). Khzama ophiolite being precisely dated at 762 ± 2 Ma (Samson et al., 2004) it allows to relatively date the Aït Ahmane ophiolite with a better accuracy than proposed in our initial manuscript (ca. 760 Ma vs. 760-720 Ma), setting the age of the hydrothermal activity and vein genesis ca. 40 Ma before the beginning of the Sturtian glaciation. We corrected this age in the revised manuscript:

L. 178-213: *“Global seawater $\delta^{18}\text{O}$ is essentially controlled by seafloor hydrothermal alteration, meaning the interaction of seawater with oceanic lithosphere in hydrothermal systems^{2-4,8,9}. The ratio between high and low temperature alteration has been evoked as maintaining the seawater $\delta^{18}\text{O}$ constant through time^{4,8,9}, partial isotopic equilibration mentioned above (Figures 5 and 6) acting as a buffer. Some authors, however, argued that this ratio evolved through time due to a two-step rise of high-temperature abyssal hydrothermalism related to geodynamic changes since the Archean². Long-term changes in sea level are also evoked as having interfered in the $\delta^{18}\text{O}$ regulation by changing the continental surface exposed to weathering². Changes in sea level also influence water pressure at the bottom of the sea, regulating the depth of fluids penetration in mid ocean ridge hydrothermal systems and thus the extent of high-temperature alteration^{2,3}. The Neoproterozoic $\delta^{18}\text{O}_{\text{seawater}}$ of $-1.33 \pm 0.98 \text{ ‰}$ that we provide here is much more precise than previous estimates (Fig. 7). This $\delta^{18}\text{O}_{\text{Nseawater}}$ value is significantly higher than that of -6.4 ‰ predicted by models considering an increasing $\delta^{18}\text{O}_{\text{seawater}}$ through time². It means that the isotopic oxygen compositions of the Neoproterozoic oceans (at 760 Ma) was similar to that of the recent oceans, which is between -1.5 and $+1.8 \text{ ‰}$ ⁴⁹, and -1.4 ‰ for an ice-free planet⁵⁰. This result*

attests to a rather constant $\delta^{18}O_{\text{seawater}}$ through time, at least since the Neoproterozoic. Hence, the ratio of low- and high-temperature hydrothermal activity, ocean volume, ridge depth and global geodynamics would have been similar than in the present day, meaning that a modern tectonic-ocean system already prevailed at 760 Ma.

Finally, the $\delta^{18}O_{\text{seawater}}$ presented here can be used to better constrain the temperature of the oceans at 760 Ma when combined with the available oxygen isotope record of authigenic carbonates and cherts. Past ocean temperatures can be estimated from the isotopic fractionation between seawater and marine sediments. These sedimentary records display a general trend of increasing $\delta^{18}O$ values from the Archean to the present^{2,4,51-54}. Considering a steady $\delta^{18}O_{\text{seawater}}$ since the Archean, this $\delta^{18}O$ increase of carbonates and cherts is generally interpreted as resulting from the progressive cooling of the ancient oceans (from ~70-50 °C during the Archean)^{4,6}. By evoking the implausibility of such high temperatures, some authors proposed that the isotopic signal of these sedimentary archives could have been modified by pervasive alteration processes such as diagenesis, post-depositional interaction with pore-water or hydrothermal fluids on the seafloor^{10,55}. Nonetheless, the fact that this trend is recorded in different mineralogies (carbonate, dolostone, chert, phosphates)^{2,3,6,7} and in different isotope systems (e.g. $\delta^{30}Si^7$, $\delta^{18}O^{2,6}$) attest to the reliability of these isotopic records. Given this, our validation of a constant $\delta^{18}O_{\text{seawater}}$ indicates that the oceans were likely 15 °C to 30 °C warmer than today 760 Myr ago, on the eve of the events of life diversification that occurred at the end of the Neoproterozoic.”

“You may reveal my name to the authors.”

Jim Kasting

Anonymous

Reviewer #2 (Remarks to the Author):

“Comments on "Fossil black smoker yields oxygen isotopic composition of Neoproterozoic seawater" by Hodel et al.

This ms reports d18O analyses of 5 samples of magnetite veins from the 760-700 Ma old Bou Azzer rocks in Morocco. The ophiolitic nature of the Bou Azzer rocks is indicated by their LREE and trace elements signatures.”

R6. There was clearly a misunderstanding here. The LREE and trace elements signatures presented in the paper were used not to state the ophiolitic nature of the serpentinites but their abyssal setting as discussed in details below (R7). They constitute strong evidences of the abyssal affinity of the hydrothermalism that affected these rocks (leading to magnetite vein precipitation).

This point and modification we brought to the manuscript are further developed in R9.

“The d18O of the magnetite veins are homogeneous and are used to back calculate the d18O of the ambient hydrothermal fluid at the time of their precipitation (assuming a given magnetite-fluid oxygen isotopic fractionation law, and a given T of formation), and in a second step, the

d18O of ambient seawater (assuming a given d18O difference between average hydrothermal fluids in black smokers and seawater). The results show a d18O of seawater similar than today and bring support to previous studies and theoretical considerations showing that the d18O of seawater has not varied by more than 1-2 permil over geological times.

The topic of this paper is interesting. The question of the surface temperature of the Earth in the Precambrian is very important. This is related to the question of the oxygen isotopic composition of seawater at that time. There are few cases where the d18O of past seawater can be constrained. Thus this ms could be very well suited for publication in Nature geosciences.

However, I see a major problem in this study, which prevents its publication in Nature geosciences or in any journal, until this problem is solved. The magnetite veins are presented as if they were precipitated in a black smoker environment and would thus give access to coeval seawater. But there is NO decisive argument given in this ms for that. There is NO decisive proof that the d18O which is reconstructed is for seawater and not for a continental hydrothermal fluid. This seems to me to be a critical problem.”

R7. The studied serpentinites clearly display a strong LREE enrichment correlated with a positive Eu anomaly that is an exclusive feature of black-smoker related serpentinites (Figures 1 and 2; e.g. Marques et al., 2006; Paulick et al., 2006; Augustin et al., 2012; Andreani et al., 2014). Moreover, Aït Ahmane serpentinites hosting the veins are also As-rich, they display important amount of sulphides and Mn-rich Cr-spinel alteration rims, which are all common characteristics of the present day black smoker related serpentinites (e.g. Marques et al., 2006; Andreani et al., 2014). Furthermore, the geochemical signature of tardi-orogenic mineralizations (Oberthür et al., 2009) is significantly different from that of the serpentinites, ruling out a post-obduction setting for the genesis of magnetite veins (Figure 2).

This point and modification we brought to the manuscript are further developed in R9.

“In addition, I find it extremely embarrassing that the first author of this ms has just published in Precambrian Research in 2017 (Hodel et al. Precambrian Research 2017, 300, 151-167) a paper saying the contrary of what is said in this ms. The abstract of this paper clearly says that there are two possibilities for the origin of the magnetite veins: " (1) a continental hydrothermal system as advanced for the Co-Ni-As ores in the Bou Azzer inlier or (2) an oceanic black smoker type hydrothermal vent field on the Neoproterozoic seafloor. "

In this paper, section 5.4 clearly describes and gives several convincing arguments for the first possibility to be correct (the same is done for the second, showing that no definitive conclusion can be reached):”

"(1) Several authors (Bouabdellah et al., 2016 and references therein) consider the Co-Ni-Fe-As-(Ag-Au) ores in the Bou Azzer inlier as a product of a polyphased hydrothermal system affecting the ultramafics between 380 and 240 Ma (Gasquet et al., 2005). These mineralizations (forming quartz-carbonates hosted ores) are typically localized at the contact between the serpentinites and the late quartz-diorite that intruded the massif (650–640 Ma, Inglis et al., 2005; Fig. 1. b). More precisely, these authors attribute the Co-Ni-Fe-arsenide mineralizations, to an intermediate phase involving Ca, Cl-rich (36–45 wt. % NaCl + CaCl equiv., Bouabdellah et al., 2016) magmatic/hydrothermal fluids, potentially mixed with meteoric water at T < 200 °C. There is nowadays a consensus on the fact that serpentinites are the sources for Co-Ni and Fe (e.g. Ahmed et al. 2009b; Bouabdellah et al., 2016). The high chlorinity reported for the

mineralizing fluids suggests a chloride complexation to explain the metals transport (e.g. Bouabdellah et al., 2016). An interaction with such fluid is denoted by the relative high chlorine concentrations in the Aït Ahmane hydrothermalized serpentinites hosting the massive magnetite veins by comparison to the unaltered ones (Figs. 6 and 13). These fluids could have mobilized the transition metals (Co, Ni and particularly Fe) in the serpentinites and precipitate Co, Ni-rich magnetite veins in cracks due to pressure drop or fluid mixing prompting the precipitation."

R8. In this same previous paper to Precambrian Research, the next (and final) part of the discussion consists in developing the hypothesis of an abyssal setting for the hydrothermal event as it follows:

“(2) An oceanic setting could also be a coherent scenario as proposed by Carbonin et al. (2015) for the Cogne (Italy) serpentinite-hosted magnetite ore. Black smoker type hydrothermal systems (e.g. Rainbow vent field) are characterized by acidic (mean pH of 2.8), hot (up to 365 °C) and Cl-rich fluids (Douville et al., 2002; Seyfried et al., 2011). These fluids are also known to be particularly enriched in Fe (and Mn, Zn, Ni, Co, Cu) due to interaction with ultramafic rocks, attesting of their ability to mobilize and transport transition metals elements (e.g. Douville et al., 2002; Marques et al., 2007). Due to Cl-complexation at low-pH, hydrothermal fluid could have leached the host serpentinite, dissolved and transported iron (and other transition metals). This is compatible with the high Cl content in serpentine phases from hydrothermalized magnetite-poor serpentinites hosting the Aït Ahmane magnetite veins (Figs. 6 and 13). Pressure drop and fluid mixing could have here again triggered the magnetite precipitation in cracks, and been responsible for the formation of massive magnetite veins of Aït Ahmane [...] Fanlo et al. (2015) suggest that such a hydrothermal vent context could explain Cr-spinels chemical particularities (high Mn and Zn contents) in the Bou Azzer ultramafics and the massive sulfides deposits in the Bou Azzer inlier (cf. Bleïda sulfides deposits). In this study, Cr-spinels rims (particularly Cr-magnetite) from hydrothermalized serpentinites hosting magnetite veins are also Mn-enriched (up to 5.41 wt% MnO; Figs. 7 and 13). Marques et al. (2007) observe similar concentrations in rims of altered Cr-spinels from the Rainbow ultramafic hosted hydrothermal field (4.9 wt% MnO). An oceanic hydrothermalism have been already mentioned by Wafik et al. (2001) in the Bou-Azzer ophiolite. The authors interpreted Cu-Fe-sulphides mineralizations in the sheeted dyke complex as indicators of a Precambrian hydrothermal activity near the paleo-spreading center, involving fluids with temperatures from 320 °C to 380 °C.”

“In the present ms, the arguments given are not decisive to my opinion (and there is no mention made of the other hypothesis described in Hodel et al. 2017). Figs 1 and 2 and the major part of the discussion on this subject relates to the serpentinites, but I think that this is not the critical point: the serpentinites have been hydrothermalized in an oceanic setting, the question is the origin of the magnetite veins. As discussed above, this can take place in a very different setting, a few 100 Ma later. Fig 3 brings argument to make the point of the authors but it is not very strong to me.”

R9. The reviewer draws the attention to a previous paper in which the two settings were discussed. In this previous paper, we concluded by saying “Concerning the Aït Ahmane serpentinites and their magnetite veins, further investigations such as a complete geochemical study of the different types of serpentinites and isotopic analysis (e.g. $\delta^{18}\text{O}$ and δD) on

magnetite veins could probably allow to discriminate between a continental and an oceanic setting”. These data, obtained subsequently to that previous work are the very results we present in the Nature Communications submission and definitively prove an abyssal setting for the magnetite veins. Furthermore, as mentioned above (R7) and contrary to what the reviewer #2 says, we mentioned the tardi-orogenic hypothesis in this present submission in the caption of the Fig. 2 saying that “Carbonates hosting tardi-orogenic mineralizations in Bou Azzer inlier, dated at $310 \pm 5 \text{ Ma}^{30}$ plot in a clearly different field (in green), discarding a post-obduction process¹¹ for Eu and LREE enrichments in the serpentinites”.

We rewrote and added the following sentences in the revised manuscript to eliminate the doubts about the abyssal affinity (black smoker type) of the hydrothermal activity responsible of the magnetite vein formation. We also modified the figures of this part (Figures 1, 2 and 3) to improve the clarity of the message that they carried.

L. 49-124: “Serpentinites from the North Aït Ahmane unit of the Bou Azzer ophiolite¹¹⁻¹⁵ (ca. $760 \text{ Ma}^{15,16}$, Morocco) experienced an intense hydrothermal activity that produced unusually massive, up to 5 cm thick magnetite veins^{11,17,18}. A detailed magneto-petrographic study¹⁸ of the hydrothermalized serpentinites hosting the veins showed that an intense iron leaching in the serpentinites by a Cl-rich acidic fluid provided the iron for magnetite precipitation. Both abyssal and tardi-orogenic settings were proposed concerning the involved hydrothermal event¹⁸. Here we provide geochemical data on the serpentinites attesting that a black smoker type (abyssal) hydrothermalism generated these unique magnetite veins.

Strong LREE and Eu enrichment are the hallmark of fluids exhaled by the present day black smoker type abyssal hydrothermal vents^{19,20} (Fig. 1 and Fig. 2). In ultramafic rocks, such REE patterns are reported only for serpentinites originated from such abyssal hydrothermal vent fields²¹⁻²⁴ (Fig. 1 and Fig. 2). Firstly interpreted as the result of fluid/rock interaction with plagioclase-bearing mafic rocks²⁵⁻²⁷, these LREE and Eu enrichments are now explained by the high mobility of these elements in acidic Cl-rich fluids, due to chlorine complexation at low pH^{19,20}. Chlorine complexation is also advanced to explain the ability of such acidic Cl-rich fluids to mobilize and transport significant amounts of transition metals, including iron (forming Fe-Cl complexes), within abyssal hydrothermal systems¹⁹. Here, we report identical geochemical features for the hydrothermalized serpentinites of the North Aït Ahmane unit (Fig. 1). These serpentinites clearly display strong LREE enrichments ($[\text{La}/\text{Yb}]_{\text{CN}}$ up to 152) correlated with positive Eu anomalies ($[\text{Eu}/\text{Eu}^*]_{\text{CN}}$ up to 27.4), contrasting with unaffected serpentinites displaying classical U-shaped REE patterns (Fig. 1 and Fig. 2). Further support for the hydrothermal origin of this chemical signature, in opposition to magmatic refertilization processes, is provided by the lack of correlation between HFSE and LREE enrichments in our samples, since HFSE are immiscible in low-temperature aqueous solutions^{21,28,29} (Fig. 3a). By contrast, the geochemical signature of carbonates related to the tardi-orogenic event³⁰ significantly differs (Fig. 2), ruling out a post-obduction setting for serpentinites alteration and magnetite veins genesis.

In addition, As and Sb concentrations also characterize serpentinites from current black smoker vent fields due to hydrothermal fluid/rock interactions^{23,24}. The high

As and Sb contents of the Aït Ahmane serpentinites (As: 0.43-224 ppm, Sb: 0.01-0.73 ppm) are akin to As and Sb contents in these modern black smokers hosted serpentinites^{23,24}. A late sedimentary origin for these high As concentrations is very unlikely given the absence of correlated LILE enrichments³¹.

Interestingly, accessory minerals are also affected in current black smoker systems, such as Cr-spinels, which are extensively altered and display important Mn-rich ferritchromite alteration rims (up to 4.53 wt.% MnO)³². As previously shown by Hodel et al.¹⁸, the hydrothermal alteration of the North Aït Ahmane serpentinites also drastically affected the Cr-spinels they host. Ferritchromite and Cr-magnetite rims resulting of this alteration are highly enriched in Mn, up to 5.41 wt.% of MnO^{18,33}, which is once again exclusive to black smoker related serpentinites³². Finally, samples presenting the highest LREE enrichments and the strongest Eu anomalies are characterized by a high abundance of sulphides and can be analogous to the sulfide-rich serpentinites and stockworks of modern black smoker systems^{23,24}.

In sum, all these petrographical and geochemical features indicate that North Aït Ahmane serpentinites endured an abyssal black smoker type hydrothermalism before the obduction of the ophiolitic sequence. Magnetite veins formation from iron leaching by acidic Cl-rich fluid¹⁸ in these serpentinites clearly results from this abyssal hydrothermalism, as further evidenced by a negative correlation between total iron content and (La/Yb)^N ratio (Fig. 3b). Thus, these massive magnetite veins and the associated hydrothermalized serpentinites likely represent the oldest fossil ultramafic-hosted black smoker type hydrothermal system ever described.”

“Otherwise all the isotopic discussion is correct, and the reconstructed d18O for the hydrothermal fluid is correct.”

R9. This statement is very important since it contradicts the main point of the reviewer #1.

“In conclusion to make it clear, it seems to me that:

- there is for sure an ophiolite at 760-700 Ma (even if it is not said how it is precisely dated)”

R10. As developed earlier in this reply, recent works of Hodel (2017) and Triantafyllou et al. (2016, 2018) also show a clear link between the Aït Ahmane ophiolite (studied here) and the Khzama ophiolite (in the Sirwa inlier ~100 km to the NW). Khzama ophiolite being precisely dated at 762 ± 2 Ma (Samson et al., 2004) it allows to relatively date the Aït Ahmane ophiolite with a better accuracy than proposed in our initial manuscript (ca. 760 Ma vs. 760-720 Ma).

“- there is for sure oceanic serpentinization with hydrothermal seawater derived fluids that caused formation of a first generation of magnetites in the serpentinites
- then there was a hydrothermal event that remobilized Fe and produced the magnetite veins. There is no decisive proof in the ms that this event was in an oceanic setting. There seems to be no proof either that it was at 700 Ma, but could have been at 380-240 Ma.”

R11. Precedently developed in R7, R8 and R9.

References included in the revised manuscript submitted to Nature Communications:

- Andreani, M., Escartin, J., Delacour, A., Ildefonse, B., Godard, M., Dymont, J., Fallick AE. and Fouquet, Y. 2014. "Tectonic Structure, Lithology, and Hydrothermal Signature of the Rainbow Massif (Mid-Atlantic Ridge 36°14'N)." *Geochemistry, Geophysics, Geosystems* 15 (9): 3543–71.
- Augustin, N., H. Paulick, K. S. Lackschewitz, A. Eisenhauer, D. Garbe-Schönberg, T. Kuhn, R. Botz, and M. Schmidt. 2012. "Alteration at the Ultramafic-Hosted Logatchev Hydrothermal Field: Constraints from Trace Element and Sr-O Isotope Data." *Geochemistry, Geophysics, Geosystems* 13.
- Block, S., J. Ganne, L. Baratoux, A. Zeh, L. A. Parra-Avila, M. Jessell, L. Ailleres, and L. Siebenaller. 2015. "Petrological and Geochronological Constraints on Lower Crust Exhumation during Paleoproterozoic (Eburnean) Orogeny, NW Ghana, West African Craton." *Journal of Metamorphic Geology* 33 (5): 463–94.
- Bodinier, J. L., C. Dupuy, and J. Dostal. 1984. "Geochemistry of Precambrian Ophiolites from Bou Azzer, Morocco." *Contributions to Mineralogy and Petrology* 87 (1). Springer-Verlag: 43–50.
- Craddock, Paul R., Wolfgang Bach, Jeffrey S. Seewald, Olivier J. Rouxel, Eoghan Reeves, and Margaret K. Tivey. 2010. "Rare Earth Element Abundances in Hydrothermal Fluids from the Manus Basin, Papua New Guinea: Indicators of Sub-Seafloor Hydrothermal Processes in Back-Arc Basins." *Geochimica et Cosmochimica Acta* 74 (19): 5494–5513.
- Douville, E., J. L. Charlou, E. H. Oelkers, P. Bienvenu, C. F. Jove Colon, J. P. Donval, Y. Fouquet, D. Prieur, and P. Appriou. 2002. "The Rainbow Vent Fluids (36°14'N, MAR): The Influence of Ultramafic Rocks and Phase Separation on Trace Metal Content in Mid-Atlantic Ridge Hydrothermal Fluids." *Chemical Geology* 184 (1–2). Elsevier: 37–48.
- Fanlo, I., F. Gervilla, V. Colás, and I. Subías. 2015. "Zn-, Mn- and Co-Rich Chromian Spinel from the Bou-Azzer Mining District (Morocco): Constraints on Their Relationship with the Mineralizing Process." *Ore Geology Reviews* 71: 82–98.
- Fein, J.B., J.J. Hemley, W.M. D'Angelo, A. Komninou, and D.A. Sverjensky. 1992. "Experimental Study of Iron-Chloride Complexing in Hydrothermal Fluids." *Geochimica et Cosmochimica Acta* 56 (8). Pergamon: 3179–90.
- Hodel, F., M. Macouin, A. Triantafyllou, J. Carlut, J. Berger, S. Rousse, N. Ennih, and R.I.F. Trindade. 2017. "Unusual Massive Magnetite Veins and Highly Altered Cr-Spinels as Relics of a Cl-Rich Acidic Hydrothermal Event in Neoproterozoic Serpentinites (Bou Azzer Ophiolite, Anti-Atlas, Morocco)." *Precambrian Research* 300.
- Hodel, F., 2017. Neoproterozoic serpentinites : a window on the oceanic lithosphere associated with the Rodinia break-up. Phd. Thesis. Université Toulouse III Paul Sabatier.
- Jaffrés, Jasmine B.D., Graham A. Shields, and Klaus Wallmann. 2007. "The Oxygen Isotope Evolution of Seawater: A Critical Review of a Long-Standing Controversy and an Improved Geological Water Cycle Model for the Past 3.4 Billion Years." *Earth-Science Reviews* 83 (1): 83–122.
- Kalczynski, Michael J., and Alexander E. Gates. 2014. "Hydrothermal Alteration, Mass Transfer and Magnetite Mineralization in Dextral Shear Zones, Western Hudson Highlands, New York, United States." *Ore Geology Reviews* 61: 226–47.
- Lanari, Pierre, Thomas Wagner, and Olivier Vidal. 2014. "A Thermodynamic Model for Di-Trioctahedral Chlorite from Experimental and Natural Data in the System MgO–FeO–Al₂O₃–SiO₂–H₂O: Applications to P–T Sections and Geothermometry." *Contributions to Mineralogy and Petrology* 167 (2): 968.

Leblanc, M. 1975. "Ophiolites Précambriennes et Gites Arséniés de Cobalt (Bou Azzer - Maroc)." Université Paris VI, Paris.

Marques, A. F. A., F. Barriga, V. Chavagnac, and Y. Fouquet. 2006. "Mineralogy, Geochemistry, and Nd Isotope Composition of the Rainbow Hydrothermal Field, Mid-Atlantic Ridge." *Mineralium Deposita* 41 (1). Springer-Verlag: 52–67.

Marques, Ana Filipa A., Fernando J.A.S. Barriga, and Steven D. Scott. 2007. "Sulfide Mineralization in an Ultramafic-Rock Hosted Seafloor Hydrothermal System: From Serpentinization to the Formation of Cu–Zn–(Co)-Rich Massive Sulfides." *Marine Geology* 245 (1): 20–39.

Naidoo, D. D., S. H. Bloomer, A. Saquaque, and K. Hefferan. 1991. "Geochemistry and Significance of Metavolcanic Rocks from the Bou Azzer-El Graara Ophiolite (Morocco)." *Precambrian Research* 53 (1–2). Elsevier: 79–97.

Oberthür, T., F. Melcher, F. Henjes-Kunst, A. Gerdes, H. Stein, A. Zimmerman, and M. El Ghorfi. 2009. "Hercynian Age of the Colbalt-Nickel-Arsenide-(Gold) Ores, Bou Azzer, Anti-Atlas, Morocco: Re-Os, Sn-Nd, and U-Pb Age Determinations." *Economic Geology* 104: 1065–79.

Paulick, H., W. Bach, M. Godard, J.C.M. De Hoog, G. Suhr, and J. Harvey. 2006. "Geochemistry of Abyssal Peridotites (Mid-Atlantic Ridge, 15°20'N, ODP Leg 209): Implications for Fluid/rock Interaction in Slow Spreading Environments." *Chemical Geology* 234 (3): 179–210.

Purtov, V. K., V. V. Kholodnov, V. N. Anfilogov, and G. S. Nechkin. 1989. "The Role of Chlorine in the Formation of Magnetite Skarns." *International Geology Review* 31 (1). Taylor & Francis Group: 63–71.

Tartèse, R., M. Chaussidon, A. Gurenko, F. Delarue, and F. Robert. 2017. "Warm Archaean Oceans Reconstructed from Oxygen Isotope Composition of Early-Life Remnants." *Geochemical Perspectives Letters*, 55–65.

Triantafyllou, A., Berger, J., Baele, J.-M., Bruguier, O., Diot, H., Ennih, N., Monnier, C., Plissart, G., Vandycke, S., Watlet, A., 2018. Intra-oceanic arc growth driven by magmatic and tectonic processes recorded in the Neoproterozoic Bougmane arc complex (Anti-Atlas, Morocco). *Precambrian Res.* 304, 39–63.

Walsh, Gregory J., Fouad Benziane, John N. Aleinikoff, Richard W. Harrison, Abdelaziz Yazidi, William C. Burton, James E. Quick, and Abderrahim Saadane. 2012. "Neoproterozoic Tectonic Evolution of the Jebel Saghro and Bou Azzer—El Graara Inliers, Eastern and Central Anti-Atlas, Morocco." *Precambrian Research* 216: 23–62.

Other references:

Carbonin, Susanna, Silvana Martin, Simone Tumati, and Piergiorgio Rossetti. 2015. "Magnetite from the Cogne Serpentinites (Piemonte Ophiolite Nappe, Italy). Insights into Seafloor Fluid–rock Interaction." *European Journal of Mineralogy* 27 (1).

Furnes, H., M. De Wit, and Y Dilek. 2014. "Four Billion Years of Ophiolites Reveal Secular Trends in Oceanic Crust Formation." *Geoscience Frontiers* 5 (4). Elsevier: 571–603.

Furnes, H., Y. Dilek, and M. de Wit. 2015. "Precambrian Greenstone Sequences Represent Different Ophiolite Types." *Gondwana Research* 27 (2): 649–85.

Seyfried, W.E., Nicholas J. Pester, Kang Ding, and Mikaella Rough. 2011. "Vent Fluid Chemistry of the Rainbow Hydrothermal System (36°N, MAR): Phase Equilibria and in Situ pH Controls on Subseafloor Alteration Processes." *Geochimica et Cosmochimica Acta* 75 (6): 1574–93.

Triantafyllou, A., Berger, J., Baele, J.-M., Diot, H., Ennih, N., Plissart, G., Monnier, C., Watlet, A., Bruguier, O., Spagna, P., Vandycke, S., 2016. The Tachakoucht–Irir–Tourtit arc complex (Moroccan Anti-Atlas): Neoproterozoic records of polyphased subduction-accretion dynamics during the Pan-African orogeny. *J. Geodyn.* 96, 81–103.

Wafik, Amina, Hassan Admou, Ali Saquaque, Abdelmajid El Boukhari, and Thierry Juteau. 2001. Cu-Fe Sulfureous Mineralisations and the Associated Alterations in the Bou Azzer and Khzama Proterozoic Op." *Ophioliti* 26 (1): 47–62.

REVIEWERS' COMMENTS:

Reviewer #1 (Remarks to the Author):

I have read through the response to reviewers, along with the revised manuscript. The responses to my own original criticisms are satisfactory. But I defer to reviewer 2 as to whether the paper deserves to be published. Reviewer 2 is much more knowledgeable about this topic than I am. His/her key question is whether the magnetite veins were really deposited coevally with the formation of the ophiolite in a midocean ridge environment during the Neoproterozoic, or whether they could have been added from hydrothermal circulation on a continent several hundred million years later. Reviewer 2 points out that first author Hodel published a paper last year (Hodel et al. Precambrian Research 2017) in which they said that there are two possibilities for the formation of the magnetite veins. If the present manuscript is published, this point needs to be made clearly so that non-experts such as myself can figure out what are the possible flaws in the argument. I realize that the present authors defend their interpretation of the veins as being midocean-ridge sourced. But I can equally well defend my assertion that their result doesn't make sense, given the proximity to the Neoproterozoic Snowball Earth glaciations. I'm perfectly content to let the resolution of this argument rest for now, as long as the reader is given a sense that the argument still exists.

Reviewer #2 (Remarks to the Author):

My major concern with the first version of this ms was that the authors did not demonstrate convincingly that the magnetite veins they used to reconstruct seawater temperature formed from a seawater hydrothermal system. Especially the authors hid in some way their previous work on this subject showing that a hydrothermal seawater origin was not the only possible one.

In the revised manuscript, the authors face this question and have a discussion relying on the REE contents of the serpentinites in which the magnetite veins developed. Their argument is based on the fact that they consider obvious that the magnetite veins developed in the same hydrothermal process that the one which imposed the REE contents of the serpentinites (and the REE of the serpentinites are consistent with a hydrothermal fluid like in a black smoker). I think this is an argument. I am not sure it is a final demonstration, especially in the case of a system which is known to have suffered hydrothermalism much later than 720 Myr ago, but at least the existing arguments are written clearly now. I still think the authors could have faced this question more in depth, using for instance minor elements in magnetite (there is a wealth of literature using minor elements in magnetites from various kinds of rocks and ore bodies to constrain their hydrothermal origin), looking perhaps to Fe isotopes, to inclusions in magnetite, ..., but I think my initial major criticism is partly released now.

Apart from that, the scientific question is important and the paper will certainly attract a real interest from the community.